# Cancer metastasis is related to normal tissue stemness

**Xing Yue Peng** **\*, Bocun Dong, Xiaohui Liu**

Biology Department, Xiamen University, Xiamen, Fujian, China

\* xypeng@xmu.edu.cn

## Abstract

The occurrence of cancer metastasis may be related to stem cells in normal tissues. We searched for patient IDs with both normal tissue stem cell values and TCGA (The Cancer Genome Atlas) clinical data for pairing and obtained 639 sets of data (stemness index of normal tissue, stemness index of tumor tissue, cancer stage, distant metastasis, tumor size) and invasion, and lymph node involvement). However, clinical data on cancer metastasis are of only four stages (e.g., Stage I, II, III, and IV), which cannot show subtle changes continuously. We need to find an effective data mining method to transform this four-valued clinical description into a numerical curve. We data-mine this data through numericalization, sorting, and noise reduction filtering. The results showed that: as the normal tissue stemness value (NS) increased, the tumor tissue stemness value (TS) increased proportionally (1.26 times NS). When NS >0.5, the rate of change in TS decelerated (0.43 times NS), and tumor metastasis began to occur. Clinical indicators, such as cancer stage, distant metastasis, tumor size and invasion, and lymph node involvement, showed that tumor metastasis became more and more severe with the increase of NS. This study suggests that tumor metastasis is triggered when the NS in the patient's body is more significant than 0.5.

## Introduction

Scientists still work hard to eliminate cancer metastasis because it is the leading cause of cancer death [1]. They attribute the cause of metastasis to microenvironment disturbance [2], intracellular gene expression [3], macrophages and leukocytes [4], hypoxia in tumor tissue [5], or fibroblasts [6]. The metastasis occurs in the lung, colon, kidney, breast, and prostate [7] through blood circulation [8] and the lymph nodes [9]. Simultaneously, surgical resection, radiotherapy, and chemotherapy are conventional clinical treatments that may accelerate cancer metastasis [10]. Scientists have concluded the paths of metastasis [11] and the circulatory system. However, less than 0.05% of tumor cells in the circulatory system survive [12]. Inside the same tumor, tumor cells' malignant potential is not the same and even [13]. The metastasis might occur incredibly early in tumorigenesis stimulated by the microenvironment [14]. Stem cell niche may promote metastasis [15] or accelerate tumor progression [16], and the microenvironmental changes by tumor growth, hypoxia, squeeze, and rupture attracts immune cells and stem cells [17]. The coculture of lung cancer cells and mesenchymal stem cells produce

**Data Availability Statement:** Relevant data is available from the figshare database (https://doi.org/10.6084/m9.figshare.20407044). Supporting information includes a database in excel format paired with the TCGA database patient ID (TCGAlong.id). Each ID contains the Stemness of

normal tissue [EREG-mRNAsi(Normal)] and the Stemness of tumor tissue [EREG-mRNAsi (Tumor1)] and their clinical cancer metastasis [transformed into numerical values, including stage (cancer), T (Tumor size), M (Distant metastases), N (Lymph node involvement)].

**Funding:** his work was supported by the National Natural Science Foundation of China (31371444). The funders had no role in study design, data collection and analysis, decision to publish, or preparation of the manuscript.

**Competing interests:** The authors have declared that no competing interests exist.

cellular spheroids [18]. Mesenchymal stem cells are abundant in animal bone marrow, and bone marrow metastasis is the most common and fatal metastasis because of stem cell induction [16]. Exosomes are extracellular vesicles that contain protein, DNA, and RNA of the cells that secrete them. They can affect cell function and cell behavior. Exosomes secreted by bone marrow mesenchymal stem cells in patients with multiple myeloma promote cancer cell growth [19]. Surgery also accumulates stem cells in the wound. Cancer metastasis has a strong relationship with inflammation [20]. Stem cells exist nearly everywhere, and stem cell accumulation in the inflammation site [21] or the wound site [22] might explain the cancer metastasis there. Stem cells may cause epithelial-mesenchymal transition [23]. It seems that the stem cells in regular tissue trigger cancer metastasis.

Currently, there is no method to continuously monitor the occurrence of cancer metastasis in vivo. This paper attempts to derive insights on when metastasis begins based on joint mining of the clinical cancer metastasis data sets with stem cell index [24] data sets representing the stem cell content of normal or tumor tissues.

## Data mining methods for clinical data

The relationship between the number of stem cells in normal tissues of cancer patients and the deterioration and metastasis of cancer first needs to be investigated because stem cells exist in normal tissues before tumor metastasis. If there are more stem cells in normal tissues, the possibility that stem cells will give rise to tumor cells is greater. If stem cells induce tumor metastasis, the data will show a positive correlation. However, the concentration of stem cells in normal human tissues cannot be directly counted. Clinical metastasis data has only four levels of qualitative description without an intermediate state. Clinical data are also mixed with various noises, such as different cancers, different organs, different ages, instrumental measurement errors and biases of experts, etc. We need to transfer clinical data into ordered arrays, filter out the noise, and get the signal (Eqs 1 and 2).

Tathiane M. Malta, et al. published the stemness index data (EREG-mRNAsi), [24] which was originally used to describe the measurement of the stemness of cancer tissues. We find a small number of normal tissue samples in these published data. We search the data for all types of tumors and have 639 stemness index measurements for normal tissues (sample type = 11) of cancer patients. In order to find out whether stem cells are related to tumor metastasis, we need to use data mining (Fig 1). We sort these data by cancer patient ID (Fig 1a), look for clinical data with the same ID in the TCGA clinical data (Fig 1b), and match each data (Fig 1c). The most important clinical data include cancer stage, tumor size and invasion, distant metastasis, and lymph node involvement, and the tags are 'Stage, T, M, N' in the TCGA database. The condition of cancer patients is represented by numbers in the TCGA database, such as I(1), II(2), III(3), IV(4). The larger the number, the more serious the condition. We use Table 1 to represent these discrete labels as 0–1 values. For example, I = 0.25, II = 0.50, III = 0.75, IV = 1.00. Taking 'Stage' as an example, Stage II, IIA, IIB are all 0.50, and Stage III, IIIA, IIIB, and IIIC are all 0.75 (see Table 1, Fig 1d).

In this way, we have numerical data on tumor progression and metastasis. The value 1.00 indicates the highest degree of cancer metastasis, and 0.25 indicates the lowest degree of cancer metastasis. Now the stemness index and clinical data are both numerical. Their distribution on the x-y scatter plot shows only four different y values (0.25, 0.50, 0.75, 1.00), representing stage I to IV of the clinical data. Such a discrete distribution is inconvenient to identify the trend of change (Fig 1e) directly. We need to sort these data according to the value of the stemness index (Fig 1e). The stem cell indexes with similar values are placed next to each other to facilitate further filtering and noise reduction by moving average (Fig 1f). Moving average is to

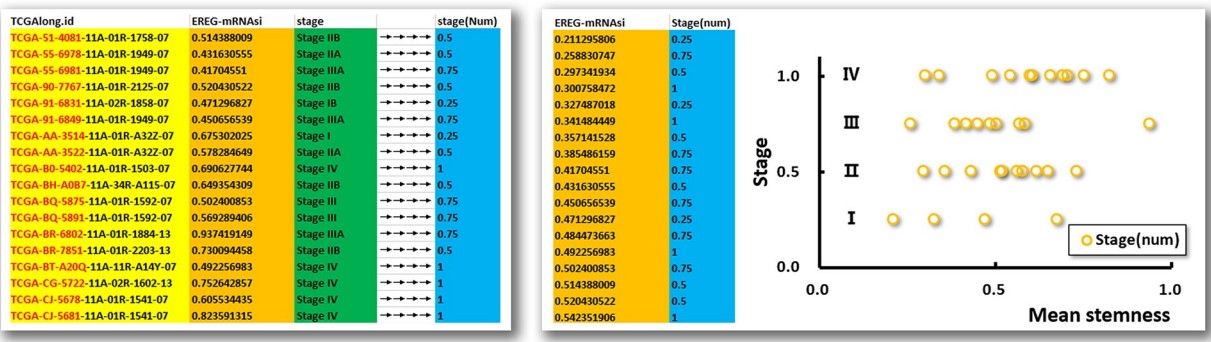

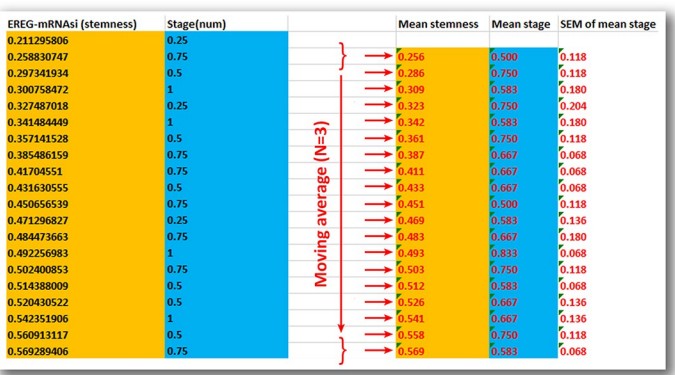
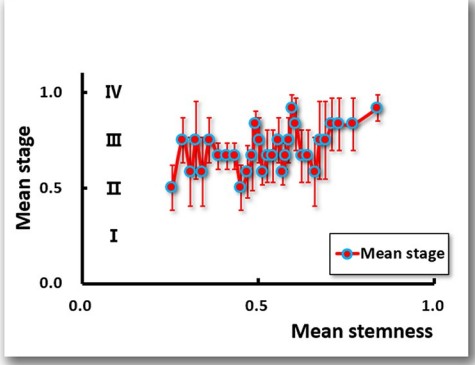

**Fig 1. The data mining processes for clinical data (stages as example).** We sort all normal solid tissue stemness data (a: EREG-mRNAsi) and clinical data (b: cancer stage) by patient IDs (TCGAlong.id or id) and pair them with the identical IDs (c). With Table 1, we give each stage a numerical value (d) and sort the data (e) by stemness index (EREG-mRNAsi). By moving the average (f: N = 3 here, N = 21 for actual data), we reduce the noise for the stage trend (g).

average the data in the window to remove the noise. The short-period noise is removed when the window moves from the minimum to the maximum (Fig 1f and 1g). The window size in Fig 1f is 3, which means that three adjacent values are averaged.

We need equitable access to all available samples with normal tissue stem cell index and clinical cancer metastasis data for actual clinical data. We first download the stemness index

**Table 1. The sample discrete value table of TCGA clinical-stage data.**

| Pathologic titles | Original labels | Given values |
|---|---|---|
| *Stage(Cancer)* | Stage I, IA, IB | 0.25 |
| *Stage(Cancer)* | Stage II, IIA, IIB | 0.5 |
| *Stage(Cancer)* | Stage III, IIIA, IIIB, IIIC | 0.75 |
| *Stage(Cancer)* | Stage IV, IVA, IVB, IVC | 1 |

data (Published by Tathiane M. Malta, et al., see S1 File, or download on https://ars.els-cdn.com/content/image/1-s2.0-S0092867418303581-mmc1.xlsx) with patient ID [24], pair the patient ID of the normal tissue stemness index (sample type = 11) with the clinical data in the TCGA database (https://portal.gdc.cancer.gov/repository) to obtain 639 sets of new data (see S1 File). We quantify the clinical data of patients according to Table 2. We now have a table of numerical "normal solid tissue stemness index (EREG-mRNAsi)", "Stage", "T", "M", "N", and "tumor stemness index".

## The moving average method for data mining

For a set of raw data $(S_i, C_i)$, we use $S_i$ to represent the stemness value and $C_i$ to the numerical clinical value $\{i = 1, 2, 3\ldots M\}$. We filter the noise of the original data $\{N < M\}$ through a moving average with a window of $N$ (a positive integer, such as $N = 3$ in Fig 1f or $N = 21$ in the calculation) to obtain a set of new data $(\alpha_j, \beta_j)$, where $\alpha_j$ is the filtered stemness data, and $\beta_j$ is the filtered clinical data $\{j = 1, 2, 3\ldots(M-N+1)\}$. The moving average process is as follows:

**Step 1**: Sort $(S_i, C_i)$ according to the value of $S_i$ to get a new order, let $S_i \leq S_{i+1}$.

**Step 2**: Take all $j$ in turn to calculate a new data set $(\alpha_j, \beta_j)$.

$$\alpha_j = \frac{\sum_{n=1}^{N}(S_{j+n-1})}{N} \tag{1}$$

$$\beta_j = \frac{\sum_{n=1}^{N}(C_{j+n-1})}{N} \tag{2}$$

**Table 2. The discrete value table of TCGA clinical data.**

| Pathologic titles | Original labels | Given values |
|---|---|---|
| *T(Tumorsize)* | T1, T1a, T1b, T1c | 0.25 |
| *T(Tumorsize)* | T2, T2a, T2b, T2c | 0.5 |
| *T(Tumorsize)* | T3, T3a, T3b, T3c | 0.75 |
| *T(Tumorsize)* | T4, T4a, T4b, T4c | 1 |
| *M(Distantmetastases)* | M0 | 0 |
| *M(Distantmetastases)* | M1 | 1 |
| *M(Distantmetastases)* | MX, unknown | |
| *N(Lymphnodeinvolvement)* | N0, N0 (i-) | 1 |
| *N(Lymphnodeinvolvement)* | N1, N1a, N1b, N1c, N1mi | 0.333 |
| *N(Lymphnodeinvolvement)* | N2, N2a, N2b, N2c | 0.667 |
| *N(Lymphnodeinvolvement)* | N3, N3a | 1 |
| *N(Lymphnodeinvolvement)* | NX, unknown | |

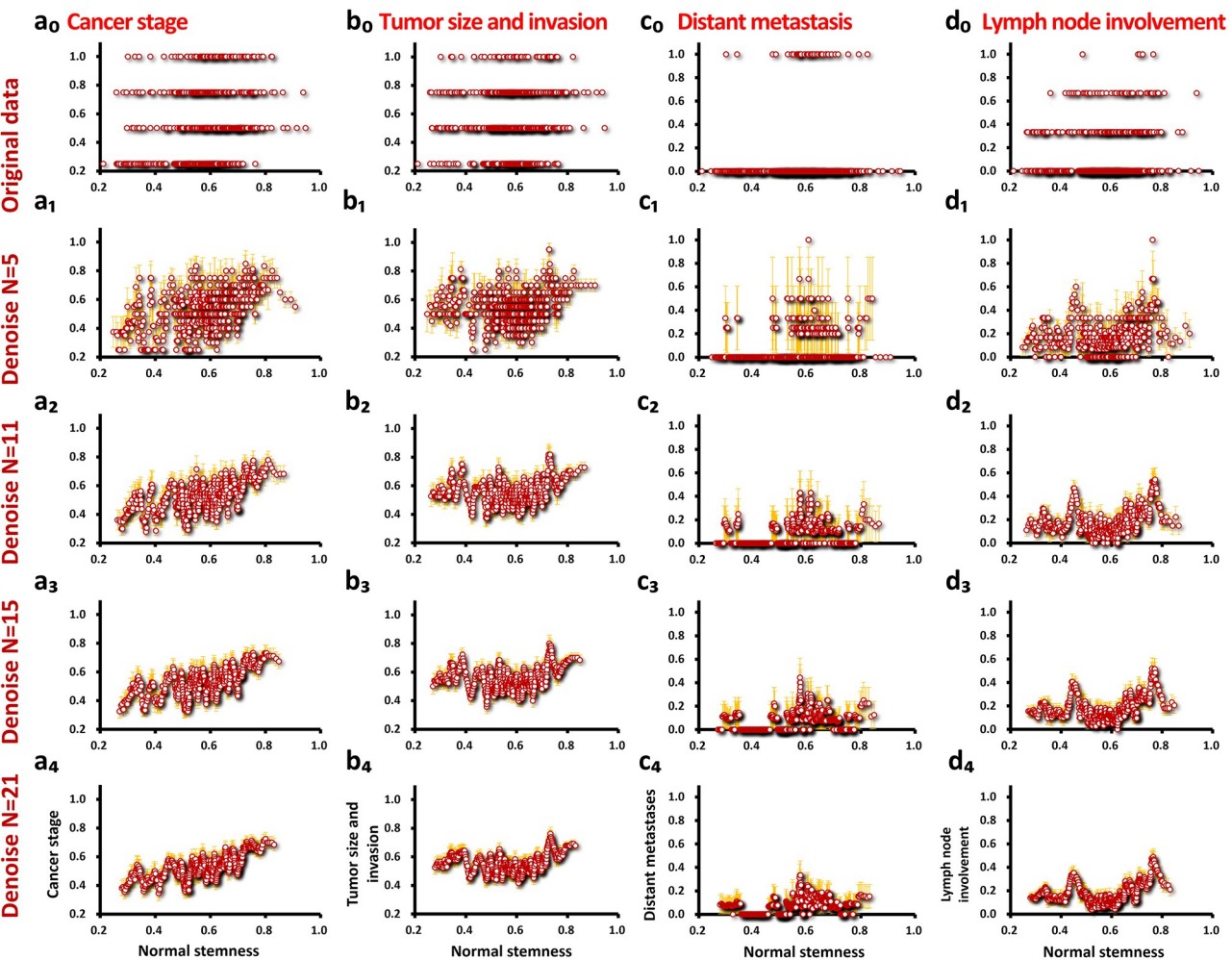

**Fig 2. The noise reduction of the four clinical data on the abscissa of the normal tissue stem cell index.** We use Table 2 to quantify the four clinical data ($a_0$, $b_0$, $c_0$ and $d_0$). Each column is the same clinical data type ($a_0$–$a_4$: cancer stage; $b_0$–$b_4$: tumor size and invasion; $c_0$–$c_4$: distant metastasis; $d_0$–$d_4$: lymph node involvement). When using higher and higher noise reduction levels (moving window $N$ in Fig 1), the noise becomes lower and lower ($N = 5$: $a_1$–$d_1$; $N = 11$: $a_2$–$d_2$; $N = 15$: $a_3$–$d_3$; $N = 21$: $a_4$–$d_4$). All abscissas are normal stemness. Orange error bars represent the SEM for each data point.

Eqs 1 and 2 are used simultaneously. When we study the relationship between normal tissue stemness index (NS: EREG-mRNAsi) and Cancer Stage (CS), we need to find all patient ids that contain both NS and CS and convert the NS and CS values of these ids into data pairs. If there are $10id_i$, $\{i = 1, 2, 3\ldots10\}$, these data pairs can be represented as $(NS_i, CS_i)$, $\{i = 1, 2, 3\ldots10\}$. We need to pair $(NS_i, CS_i)$ are sorted (Step 1) to ensure that the data pairs are ordered ($NS_i \leqslant NS_{i+1}$). If we choose $N = 3$ as the filter window, we calculate $NewNS_j$ with Eq 1 and $NewCS_j$ with Eq 2. The new data ($NewNS_j$, $NewCS_j$) is smoother and more coherent (denoised) than ($NS_i$, $CS_i$). The selection of the filter window $N$ must match the period of the noise, not the bigger, the better; see Fig 2 and discussion for details.

## Results

### The noise reduction process of the clinical data

TCGA clinical data is qualitative data represented in symbols, and there is no way to directly denoise or perform calculations on this data. After we digitize these data (Fig $2a_0$–$2d_0$), the

data distribution is discrete on the four horizontal lines of the ordinate axis (0.25, 0.5, 0.75, and 1.00). These data contain the noise (error) inherent in qualitative clinical data and fail to reveal the detailed characteristics of cancer metastasis. When we use the noise reduction filter level $N = 5$, the short-period noise lower than five is greatly eliminated. The data (Fig $2a_1$–$1d_1$) show intermediate values and general trends (relative to 0.25, 0.5, 0.75, and 1.00). When further higher filtering levels are used ($N = 11$; $N = 15$ and $N = 21$), the data ($a_2$–$a_4$, $b_2$–$b_4$, $c_2$–$c_4$ and $d_2$–$d_4$) gradually exhibit narrower fluctuations and more continuous and detailed trends. When $N = 21$ ($a_4$, $b_4$, $c_4$ and $d_4$), the changes brought about by noise reduction have slowed down. To avoid the reduction of data points and the loss of information caused by excessive noise reduction, we will use noise reduction $N = 21$ levels were obtained for further analysis ($a_4$; $b_4$; $c_4$ and $d_4$).

## The clinical data changes with normal stemness

The stemness index is obtained by regression with stem cell samples (ESC and iPSC) to show the values of cancer stemness and intratumor heterogeneity [24]. We find 639 clinical data out of 9628 with normal tissue stemness index value (type 11) in TCGA (The Cancer Genome Atlas) database. The cancer stage is a comprehensive indicator commonly used in clinical practice, and it describes the size of a tumor and how far it has spread from where it originated. It includes stage 0 (the cancer is where it started (in situ) and has not spread), stage 1(the cancer is small and has not spread anywhere else), stage 2(cancer has grown but has not spread), stage 3(the cancer is larger and may have spread) and stage 4(cancer has spread to at least one other body organ, also known as "metastatic" cancer. Another system is called the TNM staging system. $T$ describes the size of the tumor (1 for small, 4 for large). $N$ stands for lymph nodes (0 means no lymph nodes have cancer, 3 means many do). $M$ stands for metastases (0 means it has not spread, 1 means it has).

The clinical data of these four indicators (Fig 3a–3d) were positively correlated with the normal tissue stemness index value (referred to as normal stemness) (*$p < 0.0001$). It can also be seen from the graph that regions of higher normal stemness (>0.5) have higher slopes (Fig 3a, 3b, and 3d, *$p < 0.0001$). Although the correlation of distant metastases in the interval (0.5, 1) is not significant, the data in this interval, especially (0.5, 0.7), are significantly higher than the interval (0, 0.5). Roughly dividing these four clinical data at normal stemness 0.5, we see that the tumor metastasis becomes serious when normal stemness >0.5. When normal stemness <0.5 (cancer stage, Fig 3a), the tumor metastasis is not significant (Tumor size and invasion, Fig 3b; Distant metastasis, Fig 3c; Lymph node involvement, Fig 3d). Therefore, there is a trigger around normal stemness = 0.5; the increase of normal stemness triggers tumor metastasis. Clinical normal stemness varies from person to person, which means that tumors with high normal stemness are prone to cancer metastasis.

## The clinical data as a function of tumor stemness

After all, stem cells must enter the tumor tissue to interact with tumor cells. The general trend of the clinical data obtained by Tumor stemness representing the number of stem cells in the tumor tissue (except for the distant metastasis) is like that of normal stemness (Fig 1a, 1b, and 1d), except that the value of the slope is small. Compared with the slope of the cancer stage with the normal stemness interval (0.5, 1)(0.7455*±0.0612), the slope of the cancer stage with the tumor stemness interval (0.5, 1) was lower (0.1861*±0.0514). The other two indicators (Fig 4b and 4d) were similar [Tumor size and invasion: (0.4008*±0.0385) vs (0.4784*±0.0582); Lymph node involvement: (0.2844*±0.0612) vs. (0.9377*±0.0624)]. The trend of Distant metastasis is the opposite [slope is negative, (−0.1478*±0.0606) vs. (0.0534±0.0814)]. This

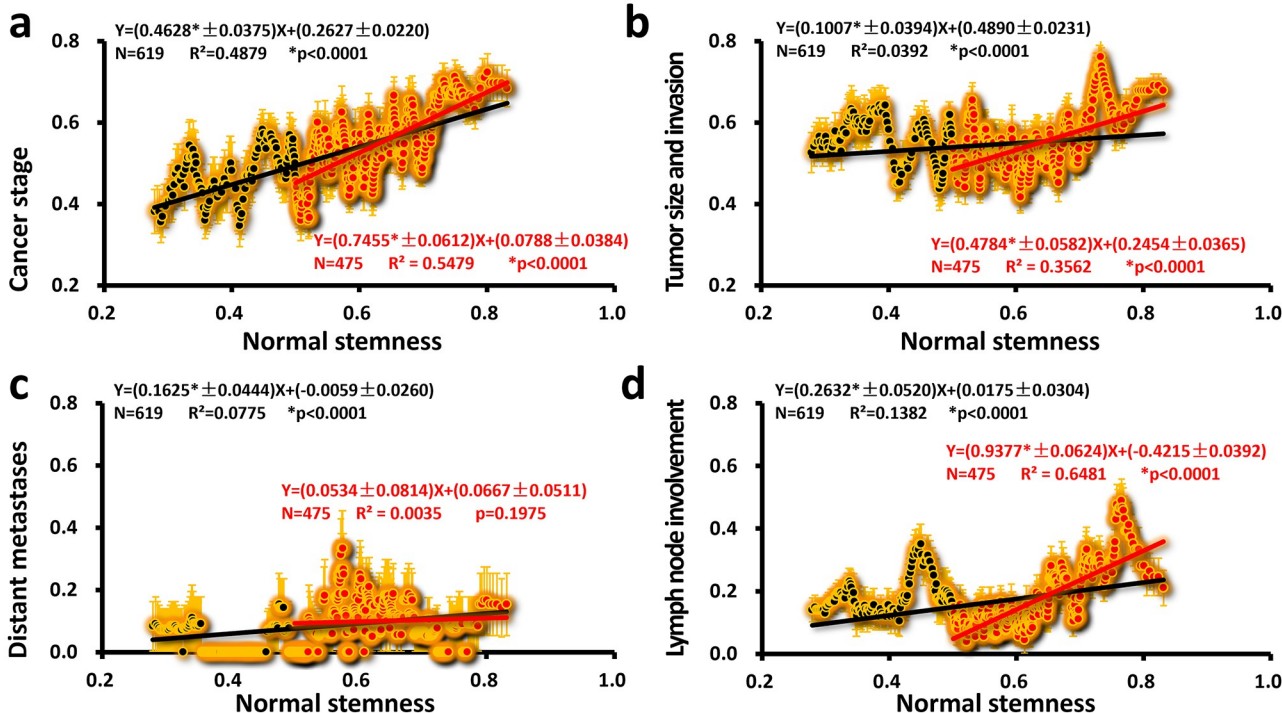

**Fig 3. Four clinical data as a function of normal stemness.** The linear regression results of the denoised clinical data (a: cancer stage; b: tumor size and invasion; c: distant metastasis; d: lymph node involvement) and Normal stemness in all intervals (0, 1) are shown as solid black lines, and black numbers represent the linear regression results on the interval (0.5, 1) with solid red lines and red numbers. Orange error bars show the calculated SEM while denoising. The error of the slope and intercept is directly expressed in the regression equation (± behind the value represents error; $N$ is the total number of data; $R^2$ is the square of the correlation coefficient; $p$ is the $p$-value for which the slope is not zero; $^*$ means significant).

negative correlation means that tumors with more severe distant metastasis have fewer stem cells. If stem cells and tumor cells also escape from the tumor tissue when cancer metastasis occurs, it will decrease stem cells in the tumor.

## The tumor stemness as a function of tumor stemness

The stemness value of tumor tissue and normal tissue is proportional in the normal stemness interval (0, 0.5) (Fig 5). The intercept (0, 0.0121±0.0414) of the regression line between them is very close to the coordinate origin (0, 0), and the slope (1.2640±0.1005) has a scaling coefficient slightly larger than 1. That is to say, in general, the stem cell index of tumor tissue should be described by the following formula (omitting the intercept for small values).

$$S_T = 1.26 S_N. \tag{3}$$

Where $S_T$ is tumor stemness, and $S_N$ is normal stemness.

This value (1.26) maybe because the tumor is slightly denser than normal tissue and contains more cells or stem cells for the same weight. As mentioned above, after normal stemness exceeds 0.5, with the increase of normal stemness, various clinical indicators show higher and higher characteristics of cancer metastasis (Fig 3). The ratio of stem cells in tumor tissues to those in normal tissues, higher tumor stemness corresponds to higher normal stemness. However, stem cells' activity should be considered a dynamic process in the human body. If the intratumoral stem cells within the tumor escape faster than the extratumoral normal stem cells into the tumor interior, the tumor stemness will deviate downward from the curve of Eq (3)

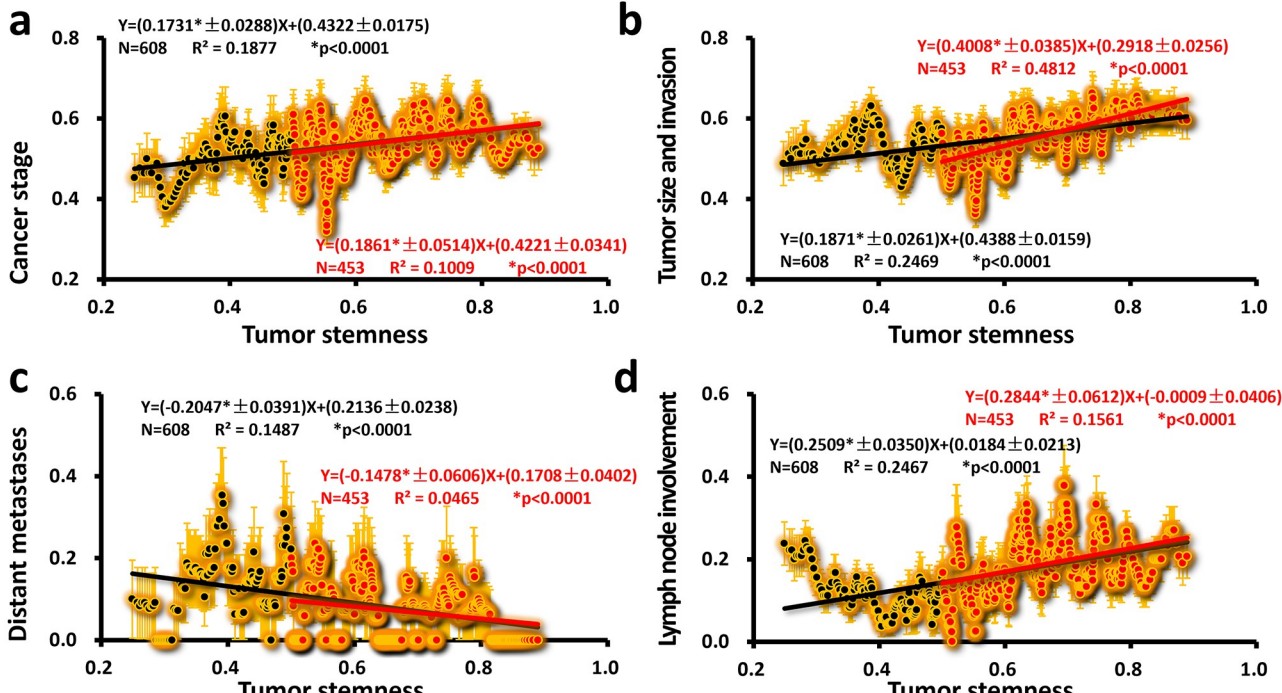

**Fig 4. Four clinical data as a function of tumor stemness.** The linear regression results of the denoised clinical data (a: cancer stage; b: tumor size and invasion; c: distant metastasis; d: lymph node involvement) and Tumor stemness in all intervals (0, 1) are shown as solid black lines, and black numbers represent the linear regression results on the interval (0.5, 1) with solid red lines and red numbers. The error in orange shows the calculated SEM while denoising. The error of the slope and intercept is directly expressed in the regression equation ± behind the value represents error; $N$ is the total number of data; $R^2$ is the square of the correlation coefficient; $p$ is the $p$-value for which the slope is not zero; * means significant).

(Fig 5, red). Assuming that the escape of stem cells and tumor cells is a synchronized phenomenon, this tumor stemness deviation is one of the characteristics of tumor metastasis (Figs 3 and 4). According to this assumption and data (Fig 5). We can compare the measured value of tumor stemness with the calculated value (Eq 3) to get the difference. This difference represents the degree of escape of stem cells in the tumor. From the figure, we can see that the maximum deviation value of this tumor stemness can reach about 0.3.

## Examples of clinicopathological pictures of different stages of different NS

The data for the 639 IDs we found were scattered across 15 tumors. With these IDs, we can directly access pathological pictures. Without data mining, the number of IDs for each tumor was small, but a positive correlation between cancer metastasis trends and NS could be seen. To avoid statistical bias, we divided the numerical interval (0−1) of NS into ten parts to form a 10 × 4 sampling grid with four metastatic stages (stage I-stage IV) (Fig 6). We sample eligible IDs and place them in the grid. If no matching ID is found, it is indicated by "/". From the occupancy of the grid, we have no ID when NS is less than 0.3. When NS gets higher and higher (0.3 < NS < 1.0), there are more and more IDs of higher Stages. This trend resembles the denoised curve of 639IDs (Fig 3a). We downloaded the pathological slides of the patient IDs of the five tumors in the lower left and upper right corners of Fig 6 (Fig 7), and compared the images with lower NS (left) and higher NS (right) in these five tumors. These pictures are not the only basis for cancer staging; they can roughly show that cancer with high NS has a higher degree of metastasis. See Table 3 for details.

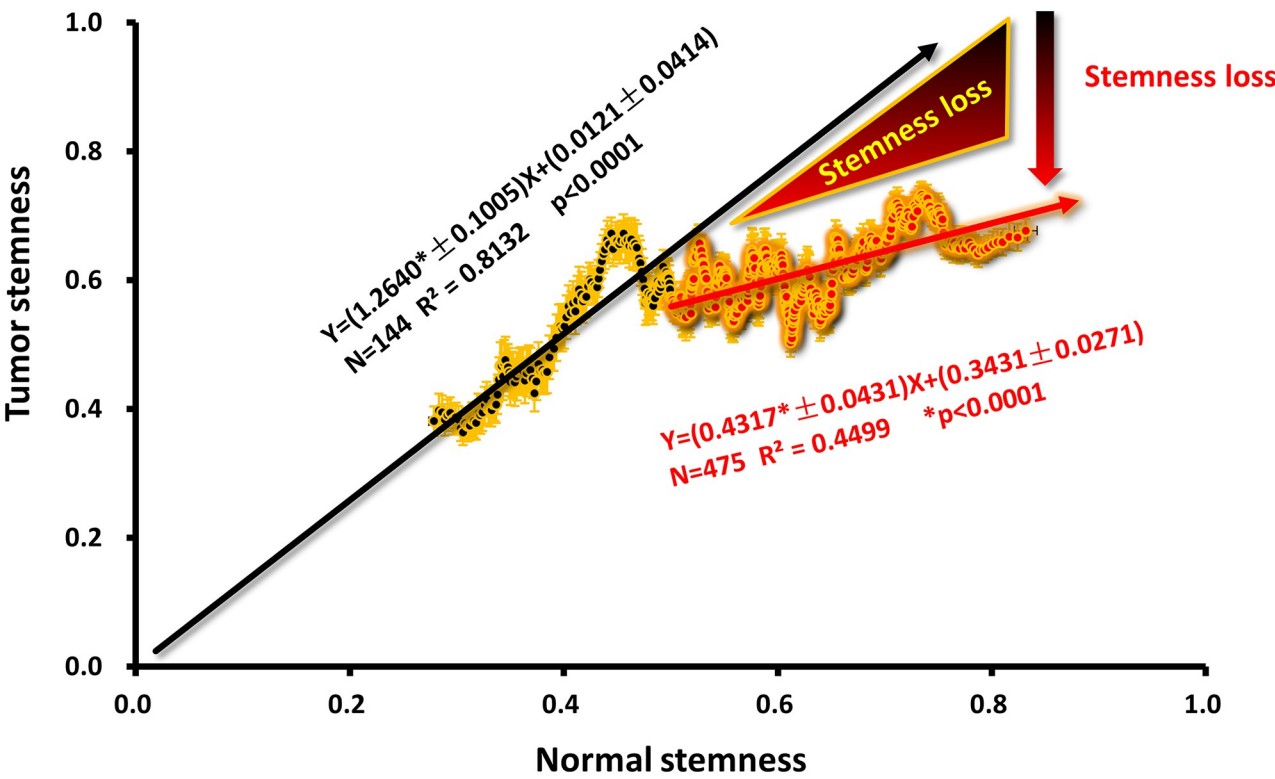

**Fig 5. Tumor stemness as a function of tumor stemness.** The linear regression results of the denoised clinical data (Tumor stemness and normal stemness) in the normal stemness interval (0, 0.5) are shown in black, and the linear regression results in the normal stemness interval (0.5, 1) are shown in red. The error in orange shows the calculated SEM while denoising. The red triangle and the black-to-red gradient arrow (labeled stemness loss) represent the difference between the measured value of Tumor stemness in the interval (0.5, 1) and the extrapolated value of the trend in the interval (0, 0.5). The error of the slope and intercept is directly expressed in the regression equation (± behind the value represents error; $N$ is the total number of data; $R^2$ is the square of the correlation coefficient; $p$ is the $p$-value for which the slope is not zero; * means significant).

## Discussion

Considering the thin layer and the permeability of capillary blood vessels, the cancer cell clusters inside the blood vessels and the stem cell clusters inside the bone marrow could be at a very close distance, less than 10μm. Any choking of the capillary blood vessels or slowing down of blood flow velocity leads to a perfect chance for stem cells and cancer cells to interact with each other and trigger metastasis. It is well known that patients with vascular disease have a 19% higher cancer risk [25] than the general population. After the surgery, the blood flow in the capillary vessels in the wounded tissue may stop. If there are still cancer cells or micro tumors left there, the probability of the cancer cell–stem cell interaction metastasis can be very high. When the tumor body grows bigger and bigger, it also presses the capillary blood vessels. It might stop the flow and accumulate stem cells and cancer cells for the high probability of the cancer cell–stem cell interaction.

Like cancer stem cells [26], the stem cells support the cancer cells for evolution, proliferation, drug resistance, cancer recurrence, and metastasis. The malignant potential of tumor cells [13] changes with the distance to the stem cells. Molecules that are similar to the ones for embryonic development [27] cause cancer cell migration because stem cells participate in this. The metastasis might occur very early [14] because the meeting of the cancer cells with the stem cells can be early or late. The environmental changes (hypoxia, squeeze, and rupture)

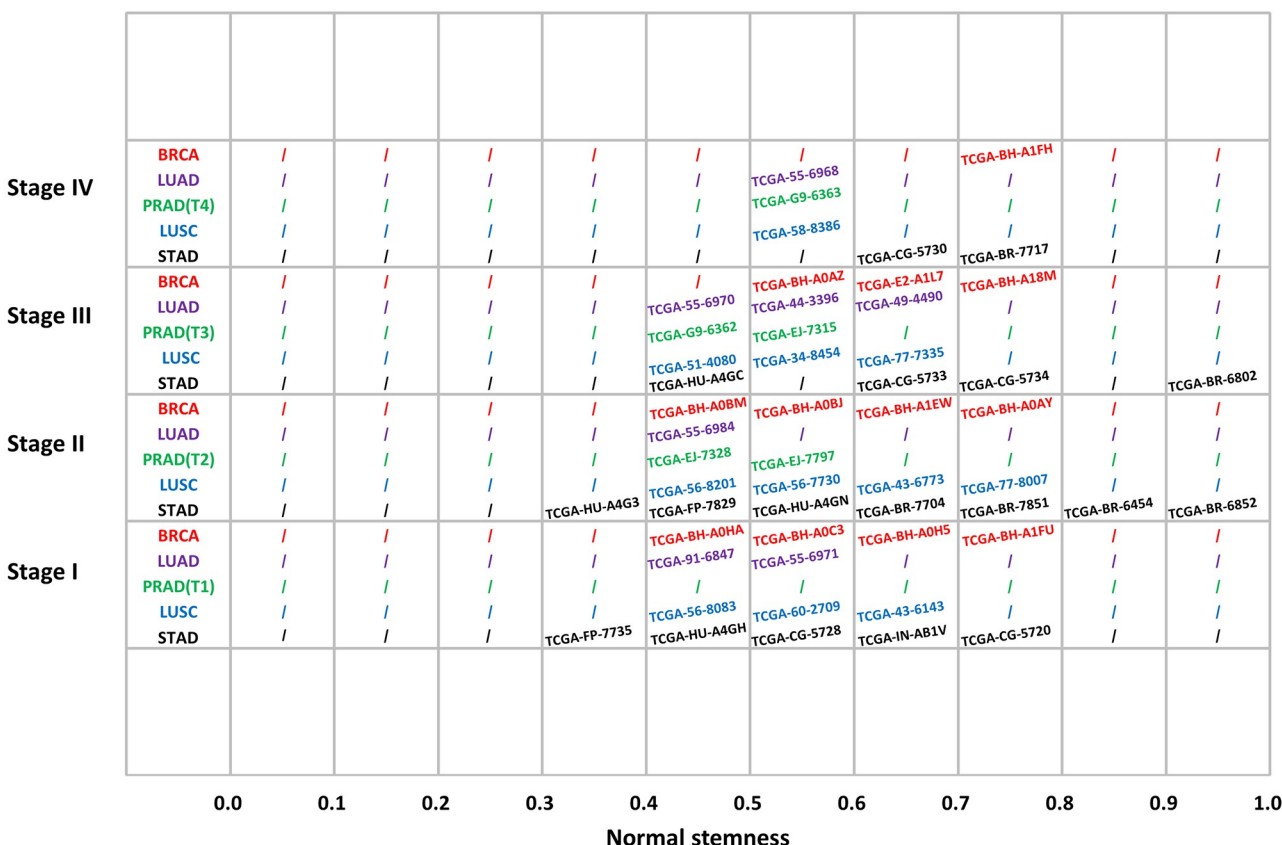

**Fig 6. TCGA Sample IDs of five tumors sampled in Normal stemness (0–1) and cancer stage (stage I–stage IV, or T1–T4) grids.** Red: BRCA (Breast invasive carcinoma); Purple: LUAD (Lung adenocarcinoma); Green: PRAD(Prostate adenocarcinoma); Blue: LUSC(Lung squamous cell carcinoma); Black: STAD(Stomach adenocarcinoma); '/': There is no eligible TCGA Sample ID in this area.

caused by tumor growth [17] attract stem cells. Bone marrow metastasis is the most common and fatal [16] because mesenchymal stem cells are abundant in the animal bone marrow. Surgical resection accelerates cancer metastasis [10] because stem cells accumulate in the wound. If there are tumor cells in the adjacent tissue where the stem cells gather, the chances of the stem cells and tumor cells being in close contact are significantly increased. If circulating tumor cells are present in the blood, stem cells and tumor cells accumulate near the wound simultaneously, resulting in close interactions between tumor cells and stem cells. Cancer metastasis has a strong relationship with inflammation [20] because stem cells are abundant there. Inflammation itself can also cause blockage of blood vessels.

We searched the entire TCGA database to obtain 639 IDs of patients tested for normal tissue stemness. Their data included cancer stage, TMN (tumor size and invasion, distant metastasis, lymph node involvement), tumor stemness, and normal stemness. The analysis after numericalization and data denoising is statistically significant. The data mining method reveals the effect of high normal stemness (NS) on tumor metastasis. Patients with higher NS (>0.5) had a higher risk of tumor progression and metastasis. This result has significant implications for the prevention and treatment of cancer metastasis. Assuming uniformity of NS throughout the human body, NS data does not need to pass a puncture sample to obtain. NS data is convenient and avoids any risky sampling near the tumor. NS is also an objective and numerical quick biochemical index with high clarity and criteria (e.g., NS>0.5). With NS's

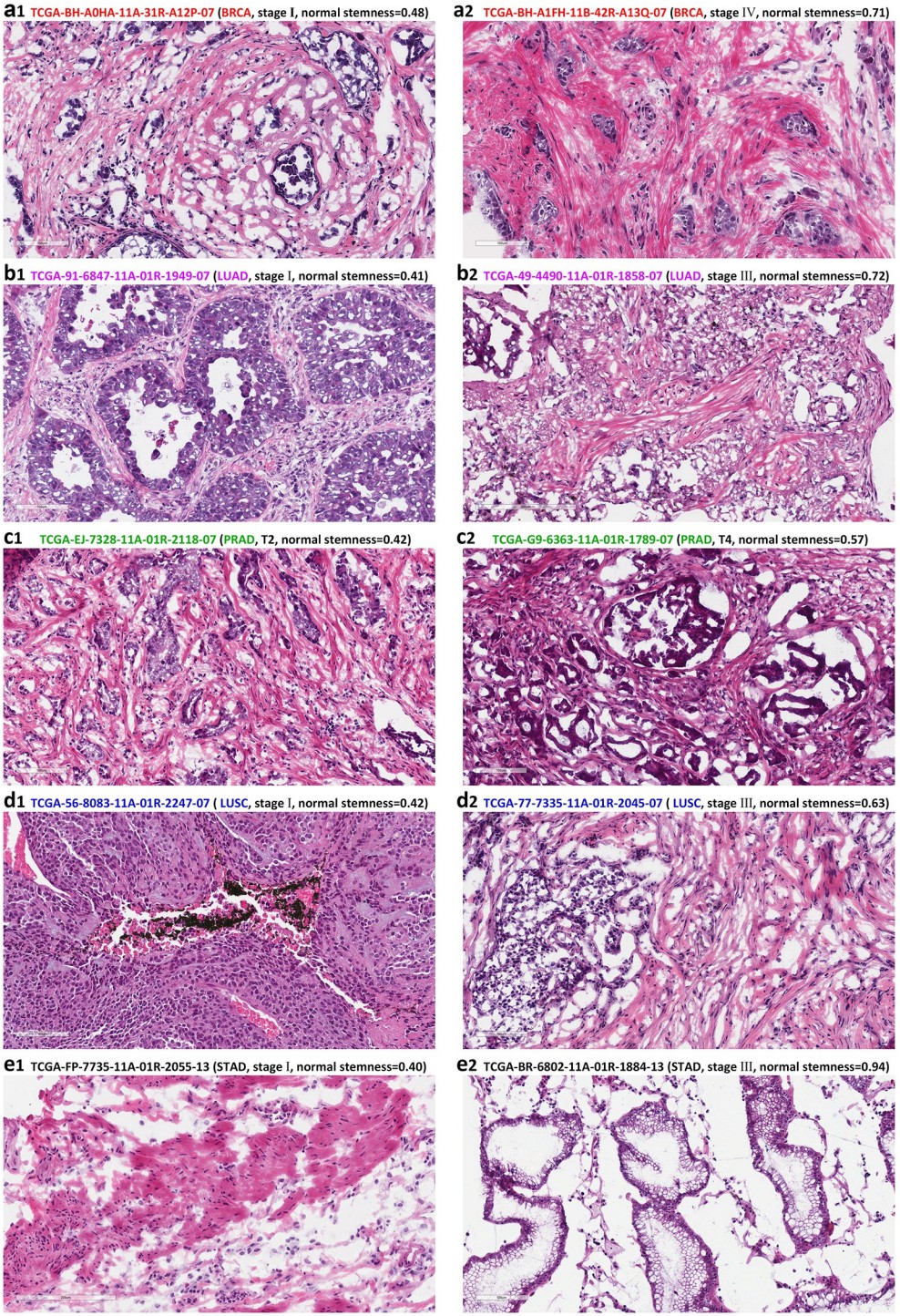

**Fig 7. Pathological section of five tumor tissues with hematoxylin and eosin staining.** Images are downloaded from the TCGA database (https://portal.gdc.cancer.gov/). IDs are taken from the lower left and upper right corners of Fig 6. The complete ID of the sample, tumor type, stage or T stage, and Normal stemness value are marked on the top of the picture. **a1–a2**: BRCA (Breast invasive carcinoma); **b1–b2**: LUAD (Lung adenocarcinoma); **c1–c2**: PRAD (Prostate adenocarcinoma); **d1–d2**: LUSC (Lung squamous cell carcinoma); **e1–e2**: STAD (Stomach adenocarcinoma). The translucent scales are at the bottom left of each image (100 μm or 200 μm).

**Table 3. The sample discrete value table of TCGA clinical-stage data.**

| Cancer type | Low NS (normal stemness) | High NS (normal stemness) |
|---|---|---|
| BRCA (Breast invasive carcinoma) | Stage I, NS = 0.48: Cancer cells are confined to the ducts, with the basement membrane intact. (Fig 7a1) | Stage IV, NS = 0.71: Cancer cells penetrate the basement membrane of breast ducts or lobules and invade the interstitium, and fibrous tissue proliferates. (Fig 7a2) |
| LUAD (Lung adenocarcinoma) | Stage I, NS = 0.41: The glandular cavity structure can be seen, the cancer cells are larger, the nucleus is larger, the degree of differentiation is higher, and there is no vascular infiltration. (Fig 7b1) | Stage III, NS = 0.72: It can be seen that tumor cells have prominent atypia, different structures, and a low degree of differentiation. Cancer cells invade surrounding blood vessels and lymph nodes. (Fig 7b2) |
| PRAD(Prostate adenocarcinoma) | T2, NS = 0.42: Small cribriform glands are seen, and the tumor is confined to the prostate capsule. (Fig 7c1) | T4, NS = 0.57: Large cribriform glands can be seen, and the tumor has exceeded the prostate capsule and invaded the seminal vesicles. (Fig 7c2) |
| LUSC(Lung squamous cell carcinoma) | Stage I, NS = 0.42: Proliferating squamous epithelial mass with keratinized cells and a high degree of differentiation can be seen. (Fig 7d1) | Stage III, NS = 0.63: Cancer cells have a low degree of differentiation and a high degree of malignancy. (Fig 7d2) |
| STAD(Stomach adenocarcinoma) | Stage I, NS = 0.40: The cancerous tissue is confined to the mucosa or submucosa. (Fig 7e1) | Stage III, NS = 0.94: Cancer tissue breaks through the mucosa or submucosa and has a low degree of differentiation. (Fig 7e2) |

fast, safe, and convenient standards, clinicians can find patients with an increased risk of tumor metastasis. Therefore, the NS index has high operability and is a promising clinical application. We can use drugs to weaken the interaction between stem cells and tumor cells or reduce the chance of stem cells and tumor cells in close contact as much as possible. It is essential to avoid blockage of blood flow. We can apply cardiovascular drugs to unclog blood vessels, lower blood lipids, or increase blood flow through physical activity. Traumatic surgery should also be applied with caution. We need to guarantee no tumor cells in the circulatory system or near the wound after surgery.

The clinical application of NS requires more significant verification in daily practice. At present, the data of clinical research on NS is minimal, and we need an extensive NS detection database to judge the NS's effectiveness. To clarify that the interaction between stem cells and tumor cells leads to tumor progression and metastasis, we need to further study the molecules, receptors and signaling pathways of this interaction. We need to find ways to block this interaction and find ways to prevent or treat cancer metastasis drug or strategy. Data mining results provide a concise experimental cell-cell metastasis model. This stem cell-tumor cell interactions in vitro model allows tumor metastasis research to have an experimental technique that mimics stem cell-triggering metastasis in vivo. Our stem cell-tumor cell experimental model provides an easily validated research platform if we want to identify the molecular mechanisms that lead to tumor metastasis. For drug development, preventing tumor metastasis can be further understood as preventing stem cells from triggering tumor metastasis. The research on drugs to prevent tumor metastasis has a clear experimental goal, and the research and development process will become more efficient.

The heterogeneity of tumor cells is a concern for stem cell-tumor cell models. At the microscopic level, tumor evolution is likely to be non-linear, and substantial genetic heterogeneity is expected in tumor cell populations [28]. Thus, the degree of tumor cell heterogeneity is an indicator of the degree of tumor progression. Cancer may contain both tumorigenic cells and non-tumorigenic cells, and their interactions can be a source of heterogeneity in evolution [29]. If the stem cells and cancer cells co-exist in the same niche, the enclosed space contains both tumorigenic and non -tumorigenic cells. If the stem cells evolve to cancer stem cells [30], it is the cancer stem cell model [31]. Biclonal evolution occurs before the multiple cells seeded metastasis [32]. Theoretically, two different cells or two subclones cannot be in the same place. The space heterogenicity may be the cause of cancer heterogenicity. The stem cells and cancer

cells offer high coevolution to metastasis [33]. Therefore, heterogeneity may be the result of the constant involvement of stem cells, and heterogeneity happens after stem cells interact with tumor cells, not before. Given the abundance of blood vessels inside tumors, and the fact that tumor growth squeezes and plugs these blood vessels, stem cell-tumor cell interactions may be insidious and early.

## Supporting information

**S1 File. Paired data.** Paired data by patient ID. 639 groups contain both normal tissue stemness and clinical cancer metastasis staging data. (https://doi.org/10.6084/m9.figshare.20407044).
(XLSX)

## Author Contributions

**Conceptualization:** Xing Yue Peng.

**Data curation:** Xing Yue Peng, Bocun Dong, Xiaohui Liu.

**Formal analysis:** Xing Yue Peng.

**Funding acquisition:** Xing Yue Peng.

**Investigation:** Xing Yue Peng.

**Methodology:** Xing Yue Peng.

**Project administration:** Xing Yue Peng.

**Resources:** Xing Yue Peng.

**Software:** Xing Yue Peng.

**Supervision:** Xing Yue Peng.

**Validation:** Xing Yue Peng.

**Visualization:** Xing Yue Peng.

**Writing – original draft:** Xing Yue Peng.

**Writing – review & editing:** Xing Yue Peng.

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
