## [Decision Letter · Decision Letter 0]

23 May 2022

PONE-D-22-09115Cancer metastasis is related to normal tissue stemnessPLOS ONE

Dear Dr. Peng,

Thank you for submitting your manuscript to PLOS ONE. After careful consideration, we feel that it has merit but does not fully meet PLOS ONE’s publication criteria as it currently stands. Therefore, we invite you to submit a revised version of the manuscript that addresses the points raised during the review process.

We look forward to receiving your revised manuscript.

Kind regards,

Filomena de Nigris, M.D., Ph.D.

Academic Editor

PLOS ONE

Journal Requirements:

Reviewers' comments:

Reviewer's Responses to Questions

**Comments to the Author**

1. Is the manuscript technically sound, and do the data support the conclusions?

Reviewer #1: Yes

Reviewer #2: Yes

2. Has the statistical analysis been performed appropriately and rigorously? 

Reviewer #1: Yes

Reviewer #2: Yes

3. Have the authors made all data underlying the findings in their manuscript fully available?

Reviewer #1: Yes

Reviewer #2: Yes

4. Is the manuscript presented in an intelligible fashion and written in standard English?

Reviewer #1: Yes

Reviewer #2: Yes

5. Review Comments to the Author

Reviewer #1: The authors correlate clinical data on cancer metastasis and stem cell index data representing stem cell content of normal or tumor tissues. Clinical data, such as cancer stage, distant metastases, tumor size and invasion, and lymph node involvement, were investigated. The results showed that tumor metastases became increasingly severe as the stemness value of normal tissue increased. Tumor metastasis is triggered when the NS in the patient's body is more significant than 0.5.

Below are my comments:

The authors provide a good method for the relationship between the number of stem cells in the normal tissues of cancer patients and cancer metastases.

1) The quality/colors of all figures needs to be Improved.

2) 2) I suggest to insert , in the discussion, the real and potential use of the NS index related to tumor metastases.

Reviewer #2: the paper entitled "Cancer metastasis is related to normal tissue stemness", besides some repeated and unclear concepts, to the simple reader, is well written and its objective is of great interest both for the Ph. D student and for the Professor. There are some points that are unclear.

The authors need to clarify, because considering the large heterogeneity of the cellular behavior of individual tumors they flatten their analysis into a single system. If the intent is to provide a system that can predict the onset of metastases in the clinical setting considering all tumors with a single behavior, I think it is wrong.

Line 9-10: clarify the concept of data-mine (for a Ph.D Student)

Line 35 : Clarify the role of exosome -

Line 37-39: Not clear the role of wound surgery in presence of stem cell.

Line 52: Clarify application and corelation of EREG-mRNAsi with line 104 / 105

Give a practical example of the validity of the proposed assumption, showing a practical clinical application example. In example: a breast / prostatic / lung tumor give from the histological analysis the possibility of metastases correlated with their real onset.

In opinion of Authors: how is possible a correlation between metastasis processes and the drug treatment phases?

6. PLOS authors have the option to publish the peer review history of their article (what does this mean?). If published, this will include your full peer review and any attached files.

Reviewer #1: No

Reviewer #2: No

---

## [Author Response · Author response to Decision Letter 0]

30 Jul 2022

-----------Questions (Q1--Q10) and answers(A1--A10)-----------------------

(Q1) The quality/colors of all figures needs to be Improved. 

(A1) See R1, R2, R3, R4. 

(Q2) I suggest to insert , in the discussion, the real and potential use of the NS index related to tumor metastases. 

(A2) See R5.

(Q3) some repeated and unclear concepts.

(A3) See R6, R9, R10, R11, R12.

(Q4) The authors need to clarify, because considering the large heterogeneity of the cellular behavior of individual tumors they flatten their analysis into a single system. If the intent is to provide a system that can predict the onset of metastases in the clinical setting considering all tumors with a single behavior, I think it is wrong. 

(A4) See R7, R8.

(Q5) Line 9-10: clarify the concept of data-mine (for a Ph.D Student)

(A5) See R9.

(Q6) Line 35 : Clarify the role of exosome –

(A6) See R10.

(Q7) Line 37-39: Not clear the role of wound surgery in presence of stem cell. 

(A7) See R11.

(Q8) Line 52: Clarify application and corelation of EREG-mRNAsi with line 104 / 105

(A8) See R12.

(Q9) Give a practical example of the validity of the proposed assumption, showing a practical clinical application example. In example: a breast / prostatic / lung tumor give from the histological analysis the possibility of metastases correlated with their real onset. 

(A9) See R13, R14, R15.

(Q10) In opinion of Authors: how is possible a correlation between metastasis processes and the drug treatment phases? 

(A10) See R16.

----------- Revisions (R1--R16)--------------------------

(R1) (for Q1) Revised Figures 1–7, all at 600dpi, 3800pt.

(R2) (for Q1) Figures 1–7 are all changed to *.png format.

(R3) (for Q1) Revised font for Figures 1–7.

(R4) (for Q1) Patched up blemishes in Figures 1–7.

(R5) (for Q2) 

The data mining results clearly show the effect of changes in stem cell index on tumor metastasis. This result has significant implications for the prevention and treatment of cancer metastasis. 

► ► ► 

The data mining method reveals the effect of high normal stemness (NS) on tumor metastasis. Patients with higher NS (>0.5) had a higher risk of tumor progression and metastasis. This result has significant implications for the prevention and treatment of cancer metastasis. Assuming uniformity of NS throughout the human body, NS data does not need to pass a puncture sample to obtain. NS data is convenient and avoids any risky sampling near the tumor. We can use drugs to weaken the interaction between stem cells and tumor cells or reduce the chance of stem cells and tumor cells in close contact as much as possible. It is essential to avoid blockage of blood flow. We can apply cardiovascular drugs to unclog blood vessels, lower blood lipids, or increase blood flow through physical activity. Traumatic surgery should also be applied with caution. We need to guarantee no tumor cells in the circulatory system or near the wound after surgery. 

(R6) (for Q3) See (Q5–Q8)

(R7) (for Q4) 

The heterogeneity of tumor cells is a concern for stem cell-tumor cell models. At the microscopic level, tumor evolution is likely to be non-linear, and substantial genetic heterogeneity is expected in tumor cell populations (28). Thus, the degree of tumor cell heterogeneity is an indicator of the degree of tumor progression. Cancer may contain both tumorigenic cells and non-tumorigenic cells, and their interactions can be a source of heterogeneity in evolution (29). If the stem cells and cancer cells co-exist in the same niche, the enclosed space contains both tumorigenic and non -tumorigenic cells. If the stem cells evolve to cancer stem cells (30), it is the cancer stem cell model (31). Biclonal evolution occurs before the multiple cells seeded metastasis (32). Theoretically, two different cells or two subclones cannot be in the same place. The space heterogenicity may be the cause of cancer heterogenicity. The stem cells and cancer cells offer high coevolution to metastasis (33). Therefore, heterogeneity may be the result of the constant involvement of stem cells, and heterogeneity happens after stem cells interact with tumor cells, not before. Given the abundance of blood vessels inside tumors, and the fact that tumor growth squeezes and plugs these blood vessels, stem cell-tumor cell interactions may be insidious and early.

(R8) (for Q4) Added references [28–33]

(R9) (for Q5) 

However, clinical data on cancer metastasis are of only four stages (e.g., Stage I, II, III, and IV), which cannot show subtle changes continuously. We need to find an effective data mining method to transform this four-valued clinical description into a numerical curve.

(R10) (for Q6) 

Exosomes are extracellular vesicles that contain protein, DNA, and RNA of the cells that secrete them. They can affect cell function and cell behavior.

(R11) (for Q7) 

Surgical resection accelerates cancer metastasis (10) because stem cells accumulate in the wound. If there are tumor cells in the adjacent tissue where the stem cells gather, the chances of the stem cells and tumor cells being in close contact are significantly increased. If circulating tumor cells are present in the blood, stem cells and tumor cells accumulate near the wound simultaneously, resulting in close interactions between tumor cells and stem cells. Cancer metastasis has a strong relationship with inflammation (20) because stem cells are abundant there. Inflammation itself can also cause blockage of blood vessels.

(R12) (for Q8) 

If stem cells induce tumor metastasis, the data will show a positive correlation. However, the concentration of stem cells in normal human tissues cannot be directly counted. Tathiane M. Malta, et al. published the stemness index data (EREG-mRNAsi), (24) which was originally used to describe the measurement of the stemness of cancer tissues. We find a small number of normal tissue samples in these published data. 

► ► ►

If stem cells induce tumor metastasis, the data will show a positive correlation. However, the concentration of stem cells in normal human tissues cannot be directly counted. Clinical metastasis data has only four levels of qualitative description without an intermediate state. Clinical data are also mixed with various noises, such as different cancers, different organs, different ages, instrumental measurement errors and biases of experts, etc. We need to transfer clinical data into ordered arrays, filter out the noise, and get the signal (Equation 1–2). Tathiane M. Malta, et al. published the stemness index data (EREG-mRNAsi), (24) which was originally used to describe the measurement of the stemness of cancer tissues. We find a small number of normal tissue samples in these published data. 

Equation 1 and Equation 2 are used simultaneously. When we study the relationship between normal tissue stemness index (NS: EREG-mRNAsi) and Cancer Stage (CS), we need to find all patient ids that contain both NS and CS and convert the NS and CS values of these ids into data pairs. If there are 10 idᵢ, {i = 1,2,3...10}, these data pairs can be represented as (NSᵢ,CSᵢ), {i = 1,2,3...10}. We need to pair ( NSᵢ, CSᵢ) are sorted (Step 1) to ensure that the data pairs are ordered (NSᵢ ≤ NSᵢ₊₁). If we choose N=3 as the filter window, we calculate NewNSⱼ with Equation 1 and NewCSⱼ with Equation 2. The new data (NewNSⱼ, NewCSⱼ) is smoother and more coherent (denoised) than (NSᵢ, CSᵢ). The selection of the filter window N must match the period of the noise, not the bigger, the better; see Figure 2 and discussion for details. 

(R13) (for Q9) Added a Section and Figure 6

Examples of clinicopathological pictures of different stages of different NS. 

The data for the 640 IDs we found were scattered across 15 tumors. With these IDs, we can directly access pathological pictures. Without data mining, the number of IDs for each tumor was small, but a positive correlation between cancer metastasis trends and NS could be seen. To avoid statistical bias, we divided the numerical interval (0–1) of NS into ten parts to form a 10x4 sampling grid with four metastatic stages (stage I–stage IV) (Figure 6). We sample eligible IDs and place them in the grid. If no matching ID is found, it is indicated by "/". From the occupancy of the grid, we have no ID when NS is less than 0.3. When NS gets higher and higher (0.3<NS<1.0), there are more and more IDs of higher Stages. This trend resembles the denoised curve of 640IDs (Figure 3a). We downloaded the pathological slides of the patient IDs of the five tumors in the lower left and upper right corners of Figure 6 (Figure 7), and compared the images with lower NS (left) and higher NS (right) in these five tumors. These pictures are not the only basis for cancer staging; they can roughly show that cancer with high NS has a higher degree of metastasis. See Table 3 for details. 

Figure 6. TCGA Sample IDs of five tumors sampled in Normal stemness (0–1) and cancer stage (stage I–stage IV, or T1–T4) grids. Red: BRCA (Breast invasive carcinoma); Purple: LUAD (Lung adenocarcinoma); Green: PRAD(Prostate adenocarcinoma); Blue: LUSC(Lung squamous cell carcinoma); Black: STAD(Stomach adenocarcinoma); '/': There is no eligible TCGA Sample ID in this area. 

(R14) (for Q9) Added Table 3

Table 3. Description of five typical tumor pathological sections (see Figure 7).

Cancer type Low NS (normal stemness) High NS (normal stemness)

BRCA (Breast invasive carcinoma) Stage Ⅰ, NS=0.48: Cancer cells are confined to the ducts, with the basement membrane intact. (Figure 7a1) Stage IV, NS=0.71: Cancer cells penetrate the basement membrane of breast ducts or lobules and invade the interstitium, and fibrous tissue proliferates. (Figure 7a2)

LUAD (Lung adenocarcinoma) Stage I, NS=0.41: The glandular cavity structure can be seen, the cancer cells are larger, the nucleus is larger, the degree of differentiation is higher, and there is no vascular infiltration. (Figure 7b1) Stage III, NS=0.72: It can be seen that tumor cells have prominent atypia, different structures, and a low degree of differentiation. Cancer cells invade surrounding blood vessels and lymph nodes. (Figure 7b2)

PRAD(Prostate adenocarcinoma) T2, NS=0.42: Small cribriform glands are seen, and the tumor is confined to the prostate capsule. (Figure 7c1) T4, NS=0.57: Large cribriform glands can be seen, and the tumor has exceeded the prostate capsule and invaded the seminal vesicles. (Figure 7c2)

LUSC(Lung squamous cell carcinoma) Stage I, NS=0.42: Proliferating squamous epithelial mass with keratinized cells and a high degree of differentiation can be seen. (Figure 7d1) Stage III, NS=0.63: Cancer cells have a low degree of differentiation and a high degree of malignancy. (Figure 7d2)

STAD(Stomach adenocarcinoma) Stage I, NS=0.40: The cancerous tissue is confined to the mucosa or submucosa. (Figure 7e1) Stage III, NS=0.94: Cancer tissue breaks through the mucosa or submucosa and has a low degree of differentiation. (Figure 7e2)

(R15) (for Q9) Added Figure 7

Figure 7. Pathological section of five tumor tissues with hematoxylin and eosin staining. Images are downloaded from the TCGA database (https://portal.gdc.cancer.gov/). IDs are taken from the lower left and upper right corners of Figure 6. The complete ID of the sample, tumor type, stage or T stage, and Normal stemness value are marked on the top of the picture. a1–a2: BRCA (Breast invasive carcinoma); b1–b2: LUAD (Lung adenocarcinoma); c1–c2: PRAD (Prostate adenocarcinoma); d1–d2: LUSC (Lung squamous cell carcinoma); e1–e2: STAD (Stomach adenocarcinoma). The translucent scales are at the bottom left of each image (100 μm or 200 μm). 

(R16) (for Q10) 

To clarify that the interaction between stem cells and tumor cells leads to tumor progression and metastasis, we need to further study the molecules, receptors and signaling pathways of this interaction. We need to find ways to block this interaction and find ways to prevent or treat cancer metastasis drug or strategy. Data mining results provide a concise experimental cell-cell metastasis model. This stem cell-tumor cell interactions in vitro model allows tumor metastasis research to have an experimental technique that mimics stem cell-triggering metastasis in vivo.

---

## [Decision Letter · Decision Letter 1]

24 Aug 2022

PONE-D-22-09115R1Cancer metastasis is related to normal tissue stemness

PLOS ONE

Dear Dr. Peng

Thank you for submitting your manuscript to PLOS ONE. After careful consideration, we feel that it has merit but does not fully meet PLOS ONE’s publication criteria as it currently stands. Therefore, we invite you to submit a revised version of the manuscript that addresses the points raised during the review process.

Editor comments

Please correct   some statements that are unclear or incomprehensible 

abstract line 8: "We data-mine this data mining.." delete "mining"last paragraph of the introduction: the statement "to conclude clinical cancer metastasis data and stem cell index data" -is not clear. Do the authors mean "to integrate ..." or better "to derive insights on when metastasis begins, based on joint mining of both data sets"?  Address any conflicts between the reviews so that it's clear which advice the authors should followProvide specific feedback from your evaluation of the manuscriptPlease ensure that your decision is justified on PLOS ONE’s publication criteria and not, for example, on novelty or perceived impact.

We look forward to receiving your revised manuscript.

Kind regards,

Filomena de Nigris, Ph.D.

Academic Editor

PLOS ONE

Journal Requirements:

Reviewers' comments:

Reviewer's Responses to Questions

**Comments to the Author**

1. If the authors have adequately addressed your comments raised in a previous round of review and you feel that this manuscript is now acceptable for publication, you may indicate that here to bypass the “Comments to the Author” section, enter your conflict of interest statement in the “Confidential to Editor” section, and submit your "Accept" recommendation.

Reviewer #1: All comments have been addressed

Reviewer #2: All comments have been addressed

2. Is the manuscript technically sound, and do the data support the conclusions?

Reviewer #1: Yes

Reviewer #2: Yes

3. Has the statistical analysis been performed appropriately and rigorously? 

Reviewer #1: Yes

Reviewer #2: Yes

4. Have the authors made all data underlying the findings in their manuscript fully available?

Reviewer #1: Yes

Reviewer #2: Yes

5. Is the manuscript presented in an intelligible fashion and written in standard English?

Reviewer #1: Yes

Reviewer #2: Yes

6. Review Comments to the Author

Reviewer #1: The authors correlate clinical data on cancer metastasis and stem cell index data representing stem cell content of normal or tumor tissues. Clinical data, such as cancer stage, distant metastases, tumor size and invasion, and lymph node involvement, were investigated. The results showed that tumor metastases became increasingly severe as the stemness value of normal tissue increased. Tumor metastasis is triggered when the NS in the patient's body is more significant than 0.5.

Overall, this is a manuscript improved by the changes made, it is now clearer. Sufficient information about the study results is presented. Methods are generally appropriate. Overall, the results were clarified.

Reviewer #2: The author responded adequately to the requests of the previous reviewers making the paper more interesting for a publication on PLOS One.

Within the discussion, the author is asked to clarify the efficiency and effectiveness of what has been identified in clinical practice.

It remains in the opinion of the undersigned that the clinical application of what has been identified requires greater verification in daily practice.

7. PLOS authors have the option to publish the peer review history of their article (what does this mean?). If published, this will include your full peer review and any attached files.

Reviewer #1: No

Reviewer #2: **Yes: **Giuseppe Palma, Isituto Nazionale Tumori -IRCCS "Fondazione G. Pascale", Naples, Italy

---

## [Author Response · Author response to Decision Letter 1]

1 Nov 2022

Dear Reviewers,

We are pleased to submit our revised manuscript (R2) entitled "Cancer metastasis is related to normal tissue stemness". 

We have carefully revised it based on editor comments and all your comments. Following all questions (Q1-Q8), we made revisions with feedback on all questions (A1-A8). We hope that this revised manuscript meets the requirements for publication.

We have submtted a manuscript entitled “A Noise Reduction Method for Discrete Clinical Data and Its Automatic Optimization Software For Fast Results” with a free software as a Lab Protocol Article. We are very happy that data miners of clinical data or researchers in other fields can save their precious time and even make new discoveries by using our software. We give it to everyone for free. 

We are very grateful to the editors and your valuable comments and suggestions.

Thank you and best regards.

Sincerely yours,

Corresponding Author

PS: -----------Questions (Q1—Q8) and answers(A1—A8)-----------------------

[Q1]: abstract line 8: "We data-mine this data mining.." delete "mining" [Editor comments]

[A1]: This word has been deleted (Abstract line 8). 

[Q2]: last paragraph of the introduction: the statement "to conclude clinical cancer metastasis data and stem cell index data" -is not clear. Do the authors mean "to integrate ..." or better "to derive insights on when metastasis begins, based on joint mining of both data sets"? [Editor comments]

[A2]: We revised to:

This paper attempts to derive insights on when metastasis begins based on joint mining of the clinical cancer metastasis data sets with stem cell index [24] data sets representing the stem cell content of normal or tumor tissues. (Line 34 … Line 35)

[Q3]: Address any conflicts between the reviews so that it's clear which advice the authors should follow [Editor comments]

[A3]: We did not find any conflicts between the reviews.

[Q4]: Provide specific feedback from your evaluation of the manuscript [Editor comments]

[A4]: The data processing methods and the research conclusions are innovative. We have submtted a manuscript entitled “A Noise Reduction Method for Discrete Clinical Data and Its Automatic Optimization Software For Fast Results” with a free software as a Lab Protocol Article (See [A6]).

[Q5]: Please ensure that your decision is justified on PLOS ONE’s publication criteria and not, for example, on novelty or perceived impact. [Editor comments]

[A5]: This article meets the publication criteria for PLOS ONE. 

[Q6]: For Lab, Study and Registered Report Protocols: These article types are not expected to include results but may include pilot data. [Editor comments] PLOS ONE offers an option for publishing peer-reviewed Lab Protocol articles, which describe protocols hosted on protocols.io. Read more information on sharing protocols at https://plos.org/protocols?utm_medium=editorial-email&utm_source=authorletters&utm_campaign=protocols. [Q6]

[A6]: We have submtted a manuscript entitled “A Noise Reduction Method for Discrete Clinical Data and Its Automatic Optimization Software For Fast Results” with a free software as a Lab Protocol Article. We are very happy that data miners of clinical data or researchers in other fields can save their precious time and even make new discoveries by using our software. We give it to everyone for free.

[Q7]: Reviewer #2: The author responded adequately to the requests of the previous reviewers making the paper more interesting for a publication on PLOS One. Within the discussion, the author is asked to clarify the efficiency and effectiveness of what has been identified in clinical practice. [Reviewer #2] 

[A7]: We further supplement the following discussion:

NS is also an objective and numerical quick biochemical index with high clarity and criteria (e.g., NS>0.5). With NS's fast, safe, and convenient standards, clinicians can find patients with an increased risk of tumor metastasis. Therefore, the NS index has high operability and is a promising clinical application. (Line 240)

[Q8]: It remains in the opinion of the undersigned that the clinical application of what has been identified requires greater verification in daily practice. [Reviewer #2] 

[A8]: We strongly agree with the reviewer's point of view. We supplement the discussion below.

The clinical application of NS requires more significant verification in daily practice. At present, the data of clinical research on NS is minimal, and we need an extensive NS detection database to judge the NS's effectiveness. (Line 247)

---

## [Editor Report · Decision Letter 2]

4 Nov 2022

Cancer metastasis is related to normal tissue stemness

PONE-D-22-09115R2

Dear Dr. Peng

We’re pleased to inform you that your manuscript has been judged scientifically suitable for publication and will be formally accepted for publication once it meets all outstanding technical requirements.

Kind regards,

Filomena de Nigris, Ph.D.

Academic Editor

PLOS ONE

---

## [Editor Report · Acceptance letter]

11 Nov 2022

PONE-D-22-09115R2 

Cancer metastasis is related to normal tissue stemness 

Dear Dr. Peng:

I'm pleased to inform you that your manuscript has been deemed suitable for publication in PLOS ONE. Congratulations! Your manuscript is now with our production department. 

Kind regards, 

on behalf of

Prof. Filomena de Nigris 

Academic Editor

PLOS ONE